# Implementation of MALDI-TOF Mass Spectrometry and Peak Analysis: Application to the Discrimination of *Cryptococcus neoformans* Species Complex and Their Interspecies Hybrids

**DOI:** 10.3390/jof6040330

**Published:** 2020-12-02

**Authors:** Margarita E. Zvezdanova, Manuel J. Arroyo, Gema Méndez, Jesús Guinea, Luis Mancera, Patricia Muñoz, Belén Rodríguez-Sánchez, Pilar Escribano

**Affiliations:** 1Clinical Microbiology and Infectious Diseases Department, Hospital General Universitario Gregorio Marañón, Doctor Esquerdo, 46, 28007 Madrid, Spain; estrellanik_zv@hotmail.es (M.E.Z.); jguineaortega@yahoo.es (J.G.); pmunoz@hggm.es (P.M.); pilar.escribano.martos@gmail.com (P.E.); 2Instituto de Investigación Sanitaria Gregorio Marañón, 28007 Madrid, Spain; 3Clover Bioanalytical Software, Centro de Empresas del Parque Tecnológico de la Salud, Av. del Conocimiento, 41, 18016 Granada, Spain; manuel.arroyo@cloverbiosoft.com (M.J.A.); gema.mendez@cloverbiosoft.com (G.M.); luis.mancera@cloverbiosoft.com (L.M.); 4CIBER de Enfermedades Respiratorias (CIBERES CB06/06/0058), 28029 Madrid, Spain; 5Medicine Department, School of Medicine, Universidad Complutense de Madrid, 28040 Madrid, Spain

**Keywords:** *Cryptococcus* spp., MALDI-TOF MS, peak analysis, in-house library, hierarchical clustering

## Abstract

Matrix-assisted laser desorption–ionization/time of flight mass spectrometry (MALDI-TOF MS) has been widely implemented for the rapid identification of microorganisms. Although most bacteria, yeasts and filamentous fungi can be accurately identified with this method, some closely related species still represent a challenge for MALDI-TOF MS. In this study, two MALDI-TOF-based approaches were applied for discrimination at the species-level of isolates belonging to the *Cryptococcus neoformans* complex, previously characterized by Amplified Fragment Length Polymorphism (AFLP) and sequencing of the ITS1-5.8S-ITS2 region: (i) an expanded database was built with 26 isolates from the main *Cryptococcus* species found in our setting (*C. neoformans*, *C. deneoformans* and AFLP3 interspecies hybrids) and (ii) peak analysis and data modeling were applied to the protein spectra of the analyzed *Cryptococcus* isolates. The implementation of the in-house database did not allow for the discrimination of the interspecies hybrids. However, the performance of peak analysis with the application of supervised classifiers (partial least squares-discriminant analysis and support vector machine) in a two-step analysis allowed for the 96.95% and 96.55% correct discrimination of *C. neoformans* from the interspecies hybrids, respectively. In addition, PCA analysis prior to support vector machine (SVM) provided 98.45% correct discrimination of the three analyzed species in a one-step analysis. This novel method is cost-efficient, rapid and user-friendly. The procedure can also be automatized for an optimized implementation in the laboratory routine.

## 1. Introduction

The genus *Cryptococcus* has classically comprised two sibling species with great clinical importance: *Cryptococcus neoformans* and *C. gattii*, the causative agents of cryptococcosis. Whilst the *C. neoformans* complex has been associated with meningitis in immunosuppressed patients, *C. gatti* has been shown to cause disease in both immune competent and immunocompromised populations [1,2,3]. Species differentiation is important in order to establish the epidemiology, virulence and susceptibility pattern to the commonly used antifungal drugs [4,5,6,7]. Traditionally, species assignment has been achieved by the morphology analysis of the colonies grown on specific culture media and serological tests [8]. The availability of DNA-based methodologies as restriction fragment length polymorphism (RFLP) analysis [9], amplified fragment length polymorphism (AFLP) analysis [10], multi-locus microsatellite typing (MLMT) [11], and multi-locus sequence typing (MLST) [12] has allowed for the identification of *Cryptococcus* species and molecular types [13,14]. Genotyping methods have allowed for the characterization of the following major molecular types: AFLP1/VNI, AFLP1A, AFLP1B/VNII for *C. neoformans*; AFLP2/VNIV for *C. deneoformans,* AFLP3/VNIII for the interspecies hybrid *C. neoformans* x *C. deneoformans*; and AFLP4/VGI, AFLP5/VGIII, AFLP6/VGII, AFLP7/VGIV and AFLP10/VGIV-VGIIIc for the *C. gattii* complex [15,16]. These molecular techniques have been shown to be accurate and robust, although the applied procedure is cumbersome, time consuming and delays the final identification. Although genomic analysis is currently the gold standard for *Cryptococcus* spp. identification, its high requirements in hands-on time and expertise has led to the evaluation of alternative tools.

Matrix-assisted laser desorption–ionization/time of flight mass spectrometry (MALDI-TOF MS) has emerged as a promising technology for the rapid and reliable identification of yeasts [17,18,19]. Isolates belonging to the *Candida* genus have been shown to be easily identified at the species level either from single colonies or directly from clinical samples using MALDI-TOF MS [20]. However, non-*Candida* yeasts still represent a challenge for this technology, especially when applied to the identification of genera poorly represented or lacking in the commercial databases [21]. In this case, expanded in-house databases containing protein spectra from the underrepresented species have been shown to overcome this drawback [17]. Although this approach has worked before for the discrimination between *C. neoformans* and *C. gatti* complexes [22,23], the construction of a database where isolates from these complexes are well represented is out of reach for most microbiology laboratories.

In this study, a novel MALDI-TOF-based methodology for the discrimination between *C. deneoformans*, *C. neoformans* and the interspecies hybrids was developed: protein spectra from *Cryptococcus* isolates were analyzed and classified using different algorithms in order to find species-specific peaks that allowed their differentiation. In addition, a database was built using well characterized isolates for the identification of the *Cryptococcus* spp. isolates using the Biotyper system developed by Bruker Daltonics (Bremen, Germany) and the identifications obtained were compared with the peak-analysis method.

## 2. Materials and Methods

### 2.1. Isolates

We retrospectively selected 70 *Cryptococcus* spp. isolates from clinical samples (*n* = 70) belonging to 67 patients admitted to Hospital Gregorio Marañón (Madrid, Spain) from 1994 to 2007 (Appendix A). Isolates were sourced from cerebrum spinal fluids (51%), blood (33%), respiratory samples (10%), and others (6%). They were morphologically identified on Columbia agar + 5% sheep blood plates (bioMérieux, Marcy-l’Étoile, France) at 35 °C, and by means of the ID 32C system (bioMérieux, Marcy-l’Étoile, France). All isolates were stored at −80 °C in water until further analysis. Twenty-six out of the seventy isolates from the study were selected randomly and included in the MALDI-TOF MS in-house library as reference main spectra profiles (MSPs). The remaining isolates (*n* = 44) were used to evaluate the commercial database alone or in combination with the in-house library. For the peak analysis approach, protein spectra from 65 *Cryptococcus* isolates were included since the remaining isolates could not be recovered from the frozen backup after MALDI-TOF MS identification.

### 2.2. Molecular Identification

To ensure the purity, the isolates were grown on Columbia agar supplemented with 5% of sheep blood plates and incubated at 35 °C for 24 h. All isolates were previously identified by the DNA sequencing analysis of the ITS1-5.8S-ITS2 region [24] and AFLP analysis [25]. Molecular identifications were considered as the reference in our study.

### 2.3. MALDI-TOF MS Identification

Seventy *Cryptococcus* spp. isolates were analyzed using an LT Microflex benchtop MALDI-TOF mass spectrometer (Bruker Daltonics, Bremen, Germany) using default settings. FlexControl 3.3 and MALDI Biotyper 3.1 were applied for spectra acquisition and comparison with the reference spectra present in the BDAL database, updated with 8223 MSPs (Bruker Daltonics). This database contains 12 reference MSPs from *C. neoformans* and 7 from *C. deneoformans*. Twenty-six isolates from the hospital collection were selected as representatives of the *Cryptococcus* species from this study and included in the in-house library (Hospital Gregorio Marañón—HGM). Then, the expanded in-house HGM library was challenged in combination with the commercial database with the remaining 44 isolates.

The sample processing method applied consisted of a mechanical disruption step followed by a standard protein extraction [26]. Briefly, a few colonies were picked, re-suspended in 300 μL water of HPLC grade (high-pressure liquid chromatography) and 900 μL ethanol, and submitted to 5 min vortexing. After a brief spin, the supernatant was discarded and the pellet allowed to dry completely at RT. Standard protein extraction with formic acid and acetonitrile was performed and 1 μL of the supernatant was spotted onto the MALDI target plate in duplicates. Once the spots were dry, they were covered with 1 μL -α-Cyano-4-hydroxycinnamic acid- (HCCA) matrix (Bruker Daltonics), prepared following the manufacturer’s instructions (Figure 1).

The identifications provided by MALDI-TOF MS were compared at the species level with those provided by AFLP analysis regardless of their score value (Table 1). Moreover, score values ≥2.0 were considered as “high-confidence” scores and those ≥1.7 as “low-confidence” ones. Score values below 1.6 were only considered as “very low-confidence” identifications—but reliable—when consistent along the four top identifications, otherwise they were considered as “not reliable”.

### 2.4. Peak Analysis

For the classification of the three *Cryptococcus* species, their protein spectra were processed using Clover MS Data Analysis software (Clover Biosoft, Granada, Spain) with the parameters shown in Appendix A in order to achieve a peak matrix with a representative mass list in the range of 2400–12,000 *m*/*z*. Furthermore, spectra alignment was performed. First, the replicates from the same isolate were aligned in order to get an average spectrum. Finally, all average spectra were aligned together.

The rate of presence for the biomarker peaks was calculated for each species and then compared among species. The receiver operating characteristic (ROC) curve with area under the curve (AUC) higher than 0.99 were used as quality indicator to measure the sensibility and specificity of a selected biomarker.

Once the putative biomarkers were selected and analyzed, a peak matrix was built containing all the aligned spectra from all *Cryptococcus* isolates, processed as described in Appendix A. This peak matrix was constructed with 10 species-specific biomarkers and it was used as input for a dendrogram obtained measuring the Euclidean distance from principal component analysis (PCA) scores.

Over the peak matrix, two approaches were applied in order to discriminate the three *Cryptococcus* species. The first one was a two-step method in which the discrimination of *C. deneoformans* from the other two species was performed as a first step and it was replicated by means of two supervised machine learning algorithms on the same peak matrix: partial least squares discriminant analysis (PLS-DA) and support vector machine (SVM). Results were validated using k-fold cross validation method.

In the second step, a new peak matrix was performed in order to achieve a better discrimination of *C. neoformans* from the interspecies hybrids. Again, PLS-DA and SVM were applied to this second peak matrix to replicate the classification and k-fold cross validation was applied. The two-step method was further improved by the removal of peaks that did not provide enough discrimination.

Finally, in order to simplify the workflow, a one-step method was assayed so that the capacity of the algorithms to discriminate the three *Cryptococcus* species at the same time was tested. In this case, only one peak matrix with spectra from the three species was built and five species-specific biomarkers were included. The alignment and processing parameters were the same as in the two-steps approach. As in the previous cases, PLS-DA and SVM analysis was performed and results were validated using k-fold confusion matrix.

### 2.5. Database Construction

Twenty-six *Cryptococcus* isolates—*C. neoformans* (*n* = 12), interspecies hybrids (*n* = 10) and *C. deneoformans* (*n* = 4)—were processed according to the manufacturer’s instructions and added to the in-house database (HGM—Hospital Gregorio Marañón library) as individual main spectra (MSPs). Following the manufacturer’s recommendations, at least ten MSPs from each species were planned to be included in the in-house library. However, this figure was modified according to the availability of isolates in our local collection.

The procedure for adding new entries to an in-house library has already been described [27]. Briefly, the instrument was calibrated before spectra acquisition using freshly prepared Bacterial Test Standard (BTS, Bruker Daltonics, Bremen, Germany); *Cryptococcus* isolates were processed as explained below and then spotted onto eight positions in the MALDI target plate and each position was read three times. Twenty-four protein spectra were thus achieved, 20 of which had to be identical in order to be accepted by the software (Biotyper, Bruker Daltonics) as an MSP and then added to the extended library.

### 2.6. Ethic Statement

The hospital Ethics Committee approved this study and gave consent for its performance (Code: MICRO.HGUGM.2017-003). Since only microbiological samples were analyzed, not human products, all the conditions to waive the informed consent were met.

## 3. Results

### 3.1. Molecular Identification

The genotyping of the isolates detected three different genotypes. The most common genotype was AFLP1/1B (*C. neoformans*, *n* = 34; 49%), followed by AFLP3 (interspecies hybrids, *n* = 29; 41%) and AFLP2 (*C. deneoformans*, *n* = 7; 10%).

### 3.2. Identification Using MALDI-TOF MS

The application of MALDI-TOF MS and the commercial database allowed the correct identification of 18/22 *C. neoformans* isolates (81.8%) and 1/3 *C. deneoformans* isolates (33.3%); the remaining *C. neoformans* isolates—*n* = 4—could not be reliably identified and for 2 *C. deneoformans* isolates, MALDI-TOF MS did not provide species identification (Table 1). The identification of the interspecies hybrids (*n* = 19) was not achieved using the commercial database due to the lack of representation of this microorganism. These isolates were identified as *C. neoformans* complex in nine cases (score ≥ 2.0, *n* = 7; score > 1.7, *n* = 1; score < 1.6, *n* = 1), as *C. deneoformans* in seven cases (score > 1.7, *n* = 4; score < 1.6, *n* = 3) and as *C. neoformans* in three cases (score > 1.7)—Table 1.

Only two *C. neoformans* isolates (4.5%) were correctly identified at the species level with high-confidence score values (≥2.0) whilst 29.5% of the samples—13—were correctly identified with low-confidence scores (≥1.7)—Table 1. Another three isolates were reliably identified to the species level, although with scores values ranging between 1.7 and 1.6—very-low confidence identifications—and finally, four isolates obtained scores below 1.6. The latter can be considered as unreliable identifications.

Using the HGM in-house library, all *C. neoformans* and *C. deneoformans* isolates were correctly identified by MALDI-TOF MS at the species level (Table 1). Moreover, 21/25 of these isolates (84.0%) were identified with score values ≥2.0 which indicates a high-confidence level. The reliability of the identification was further demonstrated by the fact that the top 4–5 identifications were identical in all cases. In all but two cases, these top reference isolates belonged to the HGM in-house library.

However, the implementation of the expanded HGM library only allowed for the correct identification of 12/19 interspecies hybrids, seven of them with score values above 2.0. The high closeness of the interspecies hybrids with the other two *Cryptococcus* species made it difficult for MALDI-TOF MS to discriminate among them and misidentified seven interspecies hybrids as *C. neoformans* (Table 1).

### 3.3. Peak Analysis

To improve the identification of the interspecies hybrids and their discrimination from *C. deneoformans* and *C. neoformans*, the peaks present in the protein spectra provided by MALDI-TOF MS were analyzed. The search for species-specific biomarker peaks yielded a list of 10 peaks that allowed the differentiation of the *Cryptococcus* species analyzed, with five of them showing higher discriminative power (Table 2 and Appendix A).

The two-step method allowed for the correct differentiation of the interspecies hybrids, which clustered distinctly in the dendrograms built using two different hierarchical clustering variations (Figure 2 and Appendix A). Accurate differentiation among the three *Cryptococcus* species was achieved using the peak matrix built upon the five most discriminative peaks, with only three spectra from interspecies hybrids misallocated in the *C. neoformans* cluster (Figure 2B). *C. neoformans* and the interspecies hybrids showed close relatedness between them based on their protein spectra. Moreover, the variability of *C. neoformans* was shown by the three main clusters in which the protein spectra from this species were grouped (Figure 2B).

The validation of the method yielded a k-fold (k = 10) score of 96.92% for PLS-DA, which performed over the peak matrix with 10 biomarkers and 98.46% for the analysis with five biomarkers. However, the SVM algorithm achieved 100% discrimination in both cases when PCA was applied (Appendix A).

A second dendrogram was performed using hierarchical clustering analysis. It showed two well defined clusters for *C. neoformans* and the interspecies hybrids (Appendix A). In this step, only three biomarkers were used to differentiate *C. deneoformans* from interspecies hybrids (5453.91, 5552.90 and 7103.00 *m*/*z*). Furthermore, this second dendrogram was validated by PLS-DA and SVM algorithms. K-fold (k = 10) was applied achieving 95.55% accuracy in both analyses.

In the single-step method, the peak matrix built with five biomarkers was used as an input for PLS-DA and SVM analysis in order to achieve the discrimination of the three *Cryptococcus* species simultaneously. PLS-DA analysis could not classify correctly the three species at the same time due to the low k-fold (k = 10) values obtained. However, the PCA performance prior to SVM allowed 98.46% correct classification of the three *Cryptococcus* species (Figure 3). The efficacy of the method was tested by k-fold (k = 10) cross validation analysis and it was above 95.0%. (Figure 3, Appendix A).

As a result of this analysis, a visual method for the differentiation of the analyzed *Cryptococcus* species can be applied based on the presence of the 6688.67 *m*/*z* peak in the *C. neoformans* isolates and its absence in *C. deneoformans* isolates, where the peaks at 6576.08 and 7103.01 *m*/*z* could be detected (Appendix A). On the other hand, both sets of peaks were present in the interspecies hybrids although some of them (2842.14, 3084.11 and 8636.24 *m*/*z*) were detected in 100% of the spectra from this species (Table 3). The visual detection of these biomarker peaks could provide a rapid and accurate identification of the *Cryptococcus* species prior to a more in-depth peak analysis using ad hoc software.

## 4. Discussion

The accurate identification of *Cryptococcus* species within the *C. neoformans* complex provides valuable information about their epidemiology, susceptibility to commonly used antifungal drugs or virulence. Our results show that discrimination among the three *Cryptococcus* species analyzed—*C. deneoformans*, *C. neoformans* and interspecies hybrids—can be performed successfully using MALDI-TOF MS and peak analysis.

The implementation of the in-house database built in our laboratory allowed for 100% correct species-level identification of the 25 *C. deneoformans* and *C. neoformans* isolates used to challenge it. Apart from the reliable identification of the analyzed *Cryptococcus* species, the in-house library also provided high confidence identifications in 63.6% of the cases. Furthermore, these results showed consistency along the 10 top identifications provided by the mass spectrometry instrument, even for the hybrids. This fact is of great importance in the routine of the microbiology laboratory in order to transfer reliable information to the clinicians.

These results are in agreement with those shown in previous studies that report improved identification of *Cryptococcus* species based on MALDI-TOF MS and the implementation of extended in-house libraries: Firacative et al. and Siqueira et al. reported 100% correct molecular-type identifications [23,27] and Posteraro et al. 98.8% unambiguous discrimination at the same level [28]. On the other hand, McTaggart et al. also showed 98.8% species-level identification although in their case no interspecies hybrids were included in the study [22]. In the case of the interspecies hybrids, Hagen et al. reported 52.0% correct identification by applying the expanded library [16]. These results match those obtained in the present study, where 61.2% (12/19) of the hybrids were reliably identified by MALDI-TOF. In both cases, the remaining isolates were identified as one of the parental species. The goal to differentiate the interspecies hybrids was only fulfilled completely in the present study when peak analysis was performed and the three *Cryptococcus* species analyzed in this study distinctively clustered together.

The updated commercial database from Bruker Daltonics has demonstrated high species-level resolution for *C. deneoformans* and *C. neoformans*—76.0%—although score values <1.7 were obtained in 21.0% of the cases and species-level identification was not provided for two *C. deneoformans* isolates. These data supported the need for expanded databases. Although the limited number of *C. deneoformans* included in this study is clearly one of its limitations, the fact that all isolates from this species were reliably identified at the species level with the expanded library shows that even small improvements in the reference library were useful to achieve the correct identifications of this species.

However, even improvements in the reference libraries proved not to be enough to differentiate the interspecies hybrids. This may be due to the algorithms used by the mass spectrometry instrument for species assignment and to the fact that the hybrids show peaks present of both parental species. Since the in-house library developed in this study has shown to be useful for the achievement of high-confidence, species-level identifications for *C. deneoformans* and *C. neoformans* and for the discrimination of 61.2% of the interspecies hybrids, the authors will be glad to share it with other users or add the reference spectra to existing open access libraries. Due to the limited number of *C. deneoformans* isolates included in our in-house library, combining our references with those from other databases could greatly improve the identification of the interspecies hybrids.

Therefore, a novel approach consisting of the analysis of the protein peaks present in the MALDI-TOF MS spectrum has been developed in this study. A list of 10 biomarker peaks was achieved as the input for the species classification. The implementation of PLS-DA analysis in a two-step approach allowed the discrimination of *C. deneoformans* isolates in the first place due to the phylogenetic differences between this species and the other two *Cryptococcus* species. Subsequently, the correct classification of *C. neoformans* isolates and the interspecies hybrids was achieved in 96.92% of the cases. Furthermore, the accuracy of this method increased when the number of biomarker peaks used was reduced to the five most discriminative ones (98.46%).

In order to simplify the analysis, a one-step method was applied to classify the three species simultaneously. In this case, PLS-DA provided correct classification in less than 75.0% of the cases but the application of SVM after PCA analysis allowed 96.92% correct discrimination of the analyzed isolates. This algorithm allowed for the correct discrimination of all *C. deneoformans* isolates and of a high rate of *C. neoformans* and interspecies hybrids despite their genetic variability, demonstrating the robustness of this study. In addition, this analysis provided a set of species-specific peaks for the *Cryptococcus* species within the *C. neoformans* complex that may be detected by visual inspection, representing a rapid and inexpensive approach for their discrimination.

## 5. Conclusions

Our results demonstrate the usefulness of MALDI-TOF MS and peak analysis when applied in the microbiology laboratory for the rapid and reliable identification of *Cryptococcus* spp. isolates. Although the updated commercial library provided correct species-level identification for a high number of *C. deneoformans* and *C. neoformans* isolates, the identification of these species was missing or not reliable in 20.5% and 18.2% of the cases, respectively. Moreover, the detection of the interspecies hybrids was not possible with the Biotyper database. However, the expanded in-house library allowed for the correct species-level identification for all *C. deneoformans* and *C. neoformans*, either by conventional identification with MALDI-TOF MS or by peak analysis. The interspecies hybrids required hierarchical clustering for their correct identification since their close relatedness with the other species made it difficult for MALDI-TOF MS to differentiate them from the other two species in a routine manner. This novel approach and the visual detection of species-specific peaks can be implemented in microbiology laboratories where ad hoc libraries are not available.

## Figures and Tables

**Figure 1 jof-06-00330-f001:**
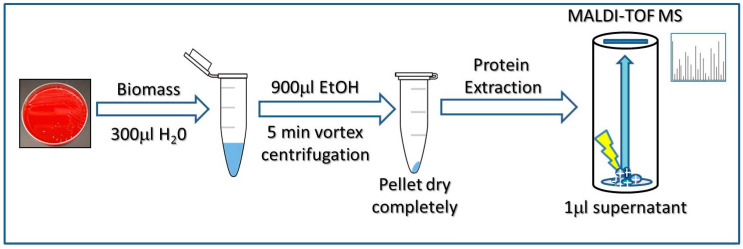
Workflow of the sample preparation method used in this study to obtain proteins from *Cryptococcus* spp. isolates for their identification by MALDI-TOF MS.

**Figure 2 jof-06-00330-f002:**
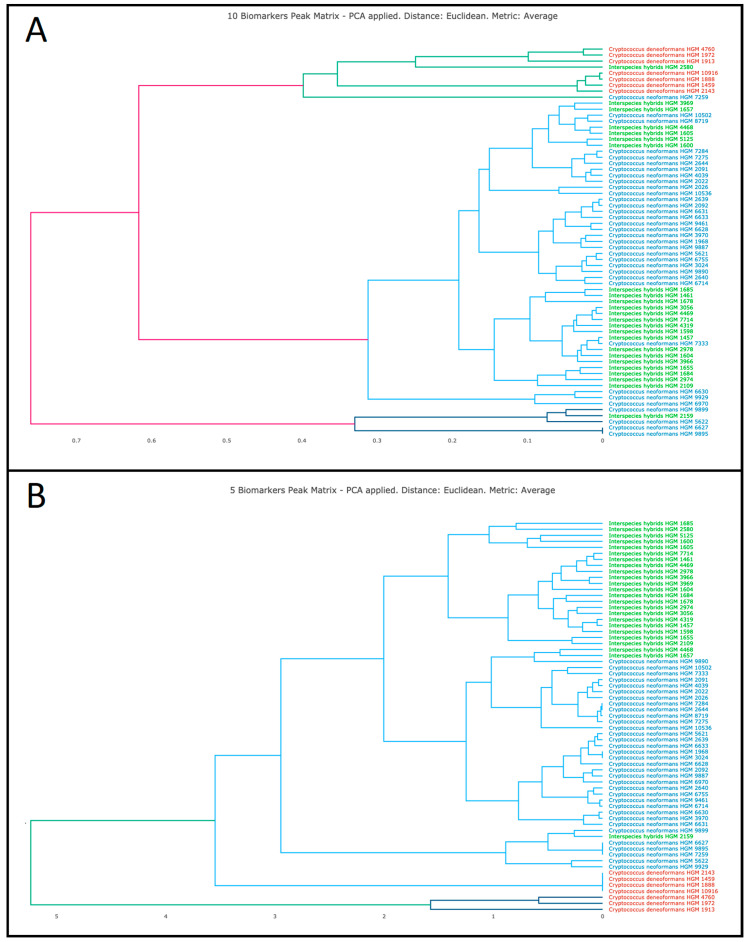
Clustering of 65 *Cryptococcus* isolates included in this study in a two-step approach. Five isolates could not be recovered from culture for further analysis: (**A**) clustering using 10 biomarker peaks and PCA; and (**B**) clustering using 5 biomarker peaks and PCA.

**Figure 3 jof-06-00330-f003:**
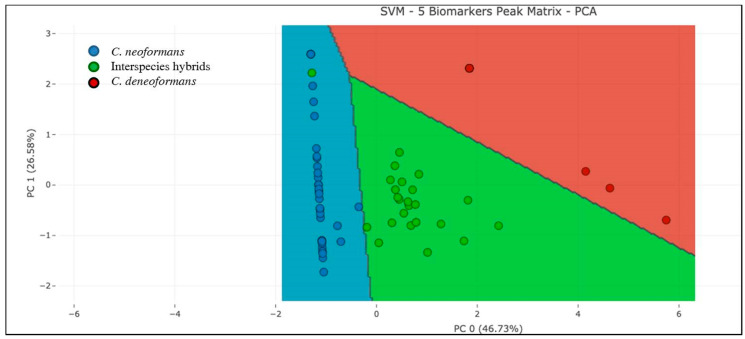
Classification of the three *Cryptococcus* species by support vector machine (SVM) in the one-step approach, using 5 biomarker peaks.

**Table 1 jof-06-00330-t001:** Identification of *Cryptococcus neoformans*, *C. deneoformans* and interspecies hybrids using MALDI-TOF MS and the Biotyper library alone or in combination with the in-house Hospital Gregorio Marañón (HGM) database. MSP stands for Main Spectra Profile.

Identification by DNA Sequencing	Isolates Analyzed	Identification Using the Biotyper Database with 8223 MSPs				Identification Using the Biotyper Database with 8223 MSPs + HGM Library		
		Score ≥ 2.0	Score ≥ 1.7	Score ≥ 1.6	Score < 1.6	Score ≥ 2.0	Score ≥ 1.7	Score ≥ 1.6
*Cryptococcus neoformans*	22	2	13	3	4	18	4	0
*Cryptococcus deneoformans*	3	0	2 ^1^	1	0	3	0	0
Interspecies hybrids	19	7 ^2^	8 ^3^	0	4 ^4^	7	12 ^5^	0
Total	44	9	23	4	8	28	16	0

^1^ Identified as *C. neoformans* complex (*n* = 2); ^2^ identified as *C. neoformans* complex (*n* = 7); ^3^ identified as *C. neoformans* complex (*n* = 1), *C. deneoformans* (*n* = 4) and *C. neoformans* (*n* = 3); ^4^ identified as *C. neoformans* complex (*n* = 1) and *C. deneoformans* (*n* = 3); ^5^ identified as *C. neoformans* (*n* = 7).

**Table 2 jof-06-00330-t002:** List of the 10 representative mass peaks of *Cryptococcus* spp. identified as potential biomarkers. These peaks were used for the construction of dendrograms and PLS-DA and SVM models. The 5 peaks marked with asterisks (*) were selected for the simplified models. CV = coefficient of variation.

*m*/*z*	Number of Spectra	*C. neoformans*	*C. neoformans* (CV)	*C. neoformans* (Mean)	Interspecies Hybrids	Interspecies Hybrids (CV)	Interspecies Hybrids (Mean)	*C. deneoformans*	*C. deneoformans* (CV)	*C. deneoformans* (mean)
2488.07	54	30/34	88.75%	4401.77	24/24	69.62%	2811.79	0/7	-	-
2842.14	53	29/34	78.21%	2202.13	24/24	79.60%	2235.20	0/7	-	-
* 3084.11	55	31/34	98.46%	7800.06	24/24	89.28%	5969.39	0/7	-	-
* 5453.91	27	1/34	0.0%	72.91	23/24	65.08%	731.90	3/7	12.65%	748.87
* 5552.90	27	1/34	0.0%	558.31	23/24	73.62%	1418.90	3/7	47.30%	2.763.28
6576.08	23	0/34	-	-	16/24	63.17%	457.98	7/7	56.70%	685.58
* 6688.67	57	34/34	95.69%	4420.91	23/24	88.39%	3556.22	0/7	-	-
* 7103.01	31	1/34	0.0%	24.32	23/24	122.76%	1484.77	7/7	52.14%	4.155.27
7830.42	18	0/34	-	-	11/24	46.49%	719.13	7/7	39.83%	449.70
8636.24	43	19/34	101.07%	2887.86	24/24	87.32%	1832.72	0/7	-	-

**Table 3 jof-06-00330-t003:** Differentiation of the analyzed *Cryptococcus* species based on the absence/presence of biomarker peaks. Figures indicate the percentage (%) of isolates showing the indicated peak.

*m*/*z*	2842.14	3084.11	6576.08	6688.67	7103.01	8636.24
*C. deneoformans*	0	0	100	0	100	0
*C. neoformans*	85.3	91.2	0	100	0	55.9
*Interspecies hybrids*	100	100	66.7	95.8	4.3	100

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
