# Peer review of "Implementation of MALDI-TOF Mass Spectrometry and Peak Analysis: Application to the Discrimination of Cryptococcus neoformans Species Complex and Their Interspecies Hybrids"

_jof, 2020, doi:10.3390/jof6040330_

Round 1
Reviewer 1 Report
Once again, the reviewer misses details and basics.
The main disadvantage of the study is the low number of isolates, apart from the fact that the manuscript is lacking accuracy.
Since this organism is not a rare one, someone could expect to include a sufficient number of strains in such a study. The database was built with 24 isolates according to the abstract, and if I am correct, according to chapter 2.5, they used 26, which number is valid?
In chapter 2.3, you state you only extracted 44 isolates. The reader is confused by the text, and it is hard to comprehend the text due to inconsistencies. The authors write them selfs that the manufacturer of the algorithm suggests at least ten isolates (L148-153) for database construction. If the authors don't have enough isolates, they have first try and get them, especially if these species are not uncommon such as many other fungi sometimes are.
What about the discussion part in which the authors put their findings against findings of previous studies that claim to have already done this, e.g., Hagen et al. 2015; Firacaiue et al. 2012 or Taggart et al. 2011?
Besides, why don't the authors make their "new" library public available, e.g. on MALDI-UP? If the results are really of much importance to the filed, this would be necessary and a real benefit to everybody.
By the way, the machine is a MALDI-TOF mass spectrometer and not only MALDI-TOF. Please change the name everywhere into MALDI-TOF MS. Which version of the Bruker Database and software did the authors use?
Table 2, for an analytic scientist, is very unusual to see such a table with more than two significant positions, e.g., 3.6 % is standard, but 3.678 % is uncommon to the reviewer. Could you explain this? The layout is not accurate. Besides, mass is not (m/z); the unit of mass would be Dalton. The authors could call it a mass peak. Since m/z means mass/charge, the real mass of a protein could be different from the detected peak.
Since it is very uncommon to take into account Score Values under 1.7, the authors have to explain to the typical MALDI-TOF MS user why they start to use a different Score Value system, as suggested by Bruker.
Please change m/z into italic. HCCA is the abbreviation; please put the abbreviation into brackets the first time you mention it.
It is hard to read Figure 2; maybe the authors show it on two pages?
The reviewer is very sorry, but in the current form, he can not support this publication.
Author Response
Once again, the reviewer misses details and basics.
A more detailed, reviewed version of the manuscript has been submitted.
The main disadvantage of the study is the low number of isolates, apart from the fact that the manuscript is lacking accuracy.
As explained above, the incidence of Cryptococcus in our hospital has been low over the last decades. However, scattered cases happen from time to time and the authors considered important to develop a rapid and reliable method for their accurate identification. The isolates included in our study were collected over a 14-year period. From our collection we could only add more C. neoformans isolates, unbalancing further the amount of isolates from this study. Getting isolates from other microbiology laboratories now, even local, is not a possibility right now as explained above. Besides, although the low number of isolates is clearly one of the study limitations, both the expanded library and peak analysis provide reliable discrimination of the three Cryptococcus species. Due to these good results, our plan is to test more isolates from different geographical and different species as soon as possible.
Since this organism is not a rare one, someone could expect to include a sufficient number of strains in such a study.
As explained above, this is not true in our hospital and it is not easy now to convince the staff from other microbiology laboratories, currently involved in COVID-19 diagnosis, to dig out their Cryptococcus isolates from their freezers.
The database was built with 24 isolates according to the abstract, and if I am correct, according to chapter 2.5, they used 26, which number is valid?
That is correct. In the abstract we had mistakenly stated that the number of isolates included in the database was 24 instead of 26. It has been corrected now. Thank you for noticing.
In chapter 2.3, you state you only extracted 44 isolates. The reader is confused by the text, and it is hard to comprehend the text due to inconsistencies.
Since the number of isolates included in the study for different purpose seems to be confusing, a paragraph has been included in Page 2, Lines 85-89: “Twenty-six out of the seventy isolates from the study were selected randomly and included in the MALDI-TOF MS in-house library as reference Main Spectra Profiles (MSPs). The remaining isolates (n=44) used to evaluate the commercial database alone or in combination with the in-house library. For the peak analysis approach, all Cryptococcus isolates from this study were included.
We do hope it is clear now. Thank you.
The authors write themselves that the manufacturer of the algorithm suggests at least ten isolates (L148-153) for database construction. If the authors don't have enough isolates, they have first try and get them, especially if these species are not uncommon such as many other fungi sometimes are.
As explained above, the limitation of 10 MSPs for accurate identifications by MALDI-TOF is only theoretical. In our laboratory, correct identifications at the species level with scores above 2.0 have been obtained with a lower number of MSPs in our in-house library. Besides, the commercial library already contains 7 MSPs for C. deneoformans. We provide 4 more MSPs and the combination of both libraries (11 MSPs in total) provide high-confidence identification for the isolates used to challenge the database (n=3). The commercial library alone only provided identifications for these 3 isolates with score values ≤ 1.7. These results support the need for 10 MSPs as recommended by the manufacturer but, as we explained before, it is not the case for other microorganisms.
What about the discussion part in which the authors put their findings against findings of previous studies that claim to have already done this, e.g., Hagen et al. 2015; Firacative et al. 2012 or Taggart et al. 2011?
Thank you for this comment. We agree that a more detailed comparison with previous studies is necessary. Therefore, this text has been included in Page 9, Lines 307-316: “These results are in agreement with those shown in previous studies that report improved identification of Cryptococcus species based on MALDI-TOF MS and the implementation of extended in-house libraries: Firacative et al. and Siqueira et al. reported 100% correct molecular-type identifications [22, 27] and Posteraro et al. 98.8% unambiguous discrimination at the same level [23]. On the other hand, McTaggart et al. also showed 98.8% species-level identification although in their case no interspecies hybrids were included in the study [21]. In the case of the interspecies hybrids, Hagen et al. reported 52.0% correct identification by applying the expanded library [15]. These results match those obtained in the present study, where 61.2% (12/19) of the hybrids were reliably identified by MALDI-TOF. In both cases, the remaining isolates were identified as one of the parental species.”
Besides, why don't the authors make their "new" library public available, e.g. on MALDI-UP? If the results are really of much importance to the filed, this would be necessary and a real benefit to everybody.
The authors have no problem whatsoever to share their library. Actually, a previous fungal library developed by the same authors is currently shared by other hospitals in Spain for its validation prior to be shared with other MALDI users.
The database can be useful to discriminate Cryptococcus species. However, as stated in the manuscript, discrimination of the Cryptococcus hybrids has not been achieved. Peak analysis, however, provides enough discrimination to differentiate the three species and the species-specific protein peaks can also be identified visually.
Anyway, the following sentence has been included in Page 10, Lines 324-328: “Since the in-house library developed in this study has shown to be useful for the achievement of high-confidence, species-level identifications for C. deneoformans and C. neoformans and for the discrimination of 61.2% of the interspecies hybrids, the authors will be glad to share it with other users or add the reference spectra to existing, open-access libraries.”
By the way, the machine is a MALDI-TOF mass spectrometer and not only MALDI-TOF. Please change the name everywhere into MALDI-TOF MS.
Done. We usually refer to it as MALDI-TOF MS but we missed “MS” in several places throughout the manuscript. We have already changed it. Thank you.
Which version of the Bruker Database and software did the authors use?
The database updated with 8223 MSPs was applied in this study (Page 3, Line 99). The software we use was FlexControl 3.3 and MALDI Biotyper 3.1. We have detailed these facts in Page 3, Lines 97-98.
Table 2, for an analytic scientist, is very unusual to see such a table with more than two significant positions, e.g., 3.6 % is standard, but 3.678 % is uncommon to the reviewer. Could you explain this? The layout is not accurate.
The calculation engine used provides 3 decimal positions. However, for the sake of consistency, two decimal positions have been shown in Table 2.
Besides, mass is not (m/z); the unit of mass would be Dalton. The authors could call it a mass peak. Since m/z means mass/charge, the real mass of a protein could be different from the detected peak.
In the MALDI-TOF methodology, singly-charged ions are primarily generated. Therefore, the mass-to-charge ratio (m/z), may be replaced directly by the monoisotopic mass of the peptides (Leopold et al., Biomolecules. 2018;8:173) However, as the reviewer rightly states, mass should be expressed in Daltons, not m/z. Thus, only m/z is shown in the header of the table.
Since it is very uncommon to take into account Score Values under 1.7, the authors have to explain to the typical MALDI-TOF MS user why they start to use a different Score Value system, as suggested by Bruker.
MALDI-users have demonstrated in several studies that the cut-off values for species-level identification can be lowered to 1.8 for fungal isolates (yeasts and molds) without decreasing the accuracy of the method (Sleiman et al., J Clin Microbiol. 2016; Schulthess et al., J Clin Microbiol. 2014, Vlek et al., J Clin Microbiol. 2014). Even a score value of 1.7 could be established as the cut-off for species level identification without affecting its reliability (Normand et al., BMC Microbiol. 2017) .The manufacturer has also lowered the score values for the identification of these microorganisms in the new versions of their software when the specific module for fungal species is applied.
Please change m/z into italic. HCCA is the abbreviation; please put the abbreviation into brackets the first time you mention it.
Done
It is hard to read Figure 2; maybe the authors show it on two pages?
We have adapted Figure 2 into a two-page format.
The reviewer is very sorry, but in the current form, he cannot support this publication.
We have made our best to make the manuscript understandable and easier to follow.
Reviewer 2 Report
The manuscript by Zvezdanova and co-workers presents the application of Matrix-Assisted Laser Desorption/Ionization Time-of Flight Mass Spectrometry and peak analysis in the Cryptococcus species identification which, as suggested by the Authors, may be used in microbiology. The presented, extensively revised version of the manuscript, fully address my previous comments. However, there are still some drawbacks listed below:
- Explain what kind of proteins did you identify, show MALDI-TOF spectra,
- check language, Moderate English changes required.
Author Response
The manuscript by Zvezdanova and co-workers presents the application of Matrix-Assisted Laser Desorption/Ionization Time-of Flight Mass Spectrometry and peak analysis in the Cryptococcus species identification which, as suggested by the Authors, may be used in microbiology. The presented, extensively revised version of the manuscript, fully address my previous comments. However, there are still some drawbacks listed below:
- Explain what kind of proteins did you identify, show MALDI-TOF spectra.
Yeast proteins ranging from 2000 to 12000 m/z are detected using the methodology described in the manuscript. Ribosomal proteins are known to be the most abundant proteins in the mass spectra but the specific identification of each of them cannot be achieved with this application (Patel, J Fungi (Basel). 2019 Jan 3;5(1):4)
Representative spectra from the three Cryptococcus species and the species-specific peaks are shown in Figure S3.
- check language, Moderate English changes required.
Done
Reviewer 3 Report
In the manuscript entitled “Implementation of MALDI-TOF Mass Spectrometry and Peak Analysis: application to the discrimination of Cryptococcus neoformans complex species and their interspecies hybrids” by Zvezdanova et al., the authors showed that MALDI-TOF MS could be successfully applied to differentiate between isolates belonging to the three major lineages in the Cryptococcus neoformans species complex: C. neoformans, C. deneoformans, and C. neoformans/C. deneoformans hybrid, by analyzing the characteristic peaks that they have identified using a collection of Cryptococcus isolates that represent the three lineages in the complex.
This finding could potentially provide a rapid and efficient way of differentiating Cryptococcus isolates that belong to different lineages, which could facilitate diagnosis and treatments in both laboratory and clinical settings. However, the study could benefit from a better presentation of the data, as well as some clarification on the data analyses and results.
Major comments
- While I appreciate that the data as presented in the study indicates MALDI-TOF MS could differentiate among the three lineages in the neoformans species complex with relatively high accuracy, especially when the peak analysis is included, one thing that I feel is missing is a list of the strains tested, and a lack of information on the genetic divergence among these strains, particularly those from the same lineage.. The conclusion would be strengthened if the authors can show that the isolates tested are good representatives of the genetic diversity in the C. neoformans species complex. There is also an underrepresentation of C. deneoformans isolates. Is there a reason for this, and would this impact the accuracy of the protocol?
- In Results 3.1, it is stated that there are a total of 70 strains that have been included in the study. However, in the subsequent sections fewer strains appear to have been analyzed by MALDI-TOF MS (44 in Table 1 and 65 in Figure 2). For example, 34 neoformans isolates were included in the “Molecular identification” section and only 22 were analyzed in the following “Identification using MALDI-TOF MS” section. If these differences in number are due to strains that were not able to be classified by the protocol described in the paper, then this should be reflected in the success rates calculated.
- Results 3.2 could be presented more clearly. For example, the beginning of line 179 reads “Only two isolates (8.0%) were …” but the identities of these two isolates are not stated, leaving the reader to guess what they are in Table 1. Also, perhaps the “23” in line 180 should be “13” instead, because if 2 isolates represent 8%, then 23 isolates should be at least >80%. Additionally, why is “Cryptococcus neoformans” bold in Table 1?
- Regarding Figures 2 and 3.
1) As mentioned previously, why are there 65 isolates in Figure 2 and only 44 isolates analyzed in Table 1?
2) The branches in Figure 2 have different colors, particularly in Figure 2A. Is there a reason for these differences?
3) What is the statistical support for the clustering in Figure 2? Can the authors use tests such as bootstrapping to evaluate the statistical significance of the branching nodes/clustering?
4) In addition to “hybrid HGM 2159”, two other hybrid isolates, “HGM 1657” and “HGM 4468”, also clearly cluster together with the other C. neoformans isolates in Figure 2B. However, in Figure 3, there appears to be only 1 hybrid isolate in the C. neoformans group.
5) Related to comment 4) above, in Figure 3, what are the criteria to divide the plot into the three areas (blue, green, and red) that represent different lineages? It seems arbitrary, and that perhaps “HGM 1657” and “HGM 4468” have been mistakenly grouped into the green area representing hybrids (based on the clustering in Figure 2B). Additionally, based on Figure 2B, the C. neoformans cluster that includes “hybrid HGM 2159” should represent a group that is more divergent from both the other hybrids as well as the rest of the C. neoformans. However, it appears that all of the C. neoformans isolates have been included in one group in Figure 3, which contradicts to the data in Figure 2B.
Minor comments
- Maybe the “complex species” in the title should be “species complex”?
- Line 55. Remove the extra “VGII” right before “for gattii”.
- Lines 83 and 84. The name of the same company is spelled differently.
- Lines 111 – 113. What is the classification for scores between 1.6 and 1.7, as in Table 1 and in line 182?
- Line 173. Should the “remaining neoformans isolates” be “remaining C. deneoformans isolates?”
- Reference No. 24. The “Cryptococcus neoformans” is missing in the title right before “species complex”.
Author Response
In the manuscript entitled “Implementation of MALDI-TOF Mass Spectrometry and Peak Analysis: application to the discrimination of Cryptococcus neoformans complex species and their interspecies hybrids” by Zvezdanova et al., the authors showed that MALDI-TOF MS could be successfully applied to differentiate between isolates belonging to the three major lineages in the Cryptococcus neoformans species complex: C. neoformans, C. deneoformans, and C. neoformans/C. deneoformans hybrid, by analyzing the characteristic peaks that they have identified using a collection of Cryptococcus isolates that represent the three lineages in the complex.
This finding could potentially provide a rapid and efficient way of differentiating Cryptococcus isolates that belong to different lineages, which could facilitate diagnosis and treatments in both laboratory and clinical settings. However, the study could benefit from a better presentation of the data, as well as some clarification on the data analyses and results.
Major comments
- While I appreciate that the data as presented in the study indicates MALDI-TOF MS could differentiate among the three lineages in the neoformans species complex with relatively high accuracy, especially when the peak analysis is included, one thing that I feel is missing is a list of the strains tested, and a lack of information on the genetic divergence among these strains, particularly those from the same lineage. The conclusion would be strengthened if the authors can show that the isolates tested are good representatives of the genetic diversity in the neoformans species complex. There is also an underrepresentation of C. deneoformans isolates. Is there a reason for this, and would this impact the accuracy of the protocol?
Information about the Cryptococcus strains analyzed has been listed in Table S1. The sample of isolation, year of isolation, their genotype and identification according to AFLP and ITS1-5.8S-ITS2 sequencing are shown. Although the number of isolates included in the study is not high, most of them (96%) sourced from different patients, and only 3 patients had two strains (patients number 5, 30 and 49) (Table S1). Therefore, we believe that the collection show high diversity, including the isolates from the same specie; in addition all isolates of C. deneoformans sourced from different patients. The low number of C. deneoformans is due to low incidence of Cryptococcus at our hospital. The molecular characterization of a high number of strains included in this study was previously published (Medical Mycology 2010, 48, 942–948, doi:10.3109/13693781003690067) showing the high genetic diversity.
In our opinion, the collection of Cryptococcus isolates included in the study are representative of its epidemiology in our geographical area. C. deneoformans is uncommonly encountered in patients from our hospital. Although this is a limitation of our study, we consider that discrimination of this species from the other two Cryptococcus species has been successfully achieved by the discovery of 3 species-specific peaks for C. deneoformans and 5 specific peaks for the other two Cryptococcus species as explained above.
This study would benefit from the further analysis of a larger number of C. deneoformans isolates and, therefore, the sentence “Due to the limited number of C. deneoformans isolates included in our in-house library, combining our references with those from other databases could greatly improved the identification of the interspecies hybrids.” has been included in Page 10, lines 328-330.
- In Results 3.1, it is stated that there are a total of 70 strains that have been included in the study. However, in the subsequent sections fewer strains appear to have been analyzed by MALDI-TOF MS (44 in Table 1 and 65 in Figure 2). For example, 34 neoformans isolates were included in the “Molecular identification” section and only 22 were analyzed in the following “Identification using MALDI-TOF MS” section. If these differences in number are due to strains that were not able to be classified by the protocol described in the paper, then this should be reflected in the success rates calculated.
For the sake of clarity, the number of isolates included in the in-house library and those used for challenging it, as well as the number of protein spectra analyzed by peak-analysis have been detailed in Page 2, Lines 85-91: “Twenty-six out of the seventy isolates from the study were selected randomly and included in the MALDI-TOF MS in-house library as reference Main Spectra Profiles (MSPs). The remaining isolates (n=44) were used to evaluate the commercial database alone or in combination with the in-house library. For the peak analysis approach, protein spectra from 65 Cryptococcus isolates were included since the remaining isolates could not be recovered from the frozen backup after MALDI-TOF MS identification.”
We do hope that this information is useful to understand how the isolates were distributed and analyzed.
- Results 3.2 could be presented more clearly. For example, the beginning of line 179 reads “Only two isolates (8.0%) were …” but the identities of these two isolates are not stated, leaving the reader to guess what they are in Table 1. Also, perhaps the “23” in line 180 should be “13” instead, because if 2 isolates represent 8%, then 23 isolates should be at least >80%. Additionally, why is “Cryptococcus neoformans” bold in Table 1?
Thank you for this comment. It will help us to explain this part more clearly. In line 186 we state now: “Only two C. neoformans isolates (8.0%) were correctly identified at the species level…”
Regarding line 180, we state “23” because this is the total number of isolates identified with score values≥1.7, but we agree that only 13 are identified correctly. The text has been changed accordingly “…whilst 29.5% of the samples -13- were identified with low-confidence scores (≥1.7) -Table 1-.”
Besides, “Cryptococcus neoformans” was in bold due to formatting issues of the table. The bold format has been removed.
- Regarding Figures 2 and 3.
1) As mentioned previously, why are there 65 isolates in Figure 2 and only 44 isolates analyzed in Table 1?
The protein spectra obtained for identification purposes were collected for peak analysis. Some of the protein spectra were good enough for identification but did not pass the quality control requested to be included in the different models shown in the study. Therefore, the spectra from several isolates were achieved again from this purpose. However, 5 interspecies hybrids could not be recovered from the frozen backup and their spectra unfortunately could not be included in the peak analysis. This fact is stated in Figure 2 legend “Five isolates could not be recovered from culture for further analysis.”
2) The branches in Figure 2 have different colors, particularly in Figure 2A. Is there a reason for these differences?
No, there is no meaning associated to the color of the branches. The software provides branches from different colors in order to differentiate clusters. The root of the dendrogram is pink in Figure 2A and 2B. The following level of branching is usually blue/navy blue when two branches are present and new colors are applied for further branches. Posteraro et al. (2012) show a similar dendrogram in their study, where Bruker Biotyper software has been applied.
3) What is the statistical support for the clustering in Figure 2? Can the authors use tests such as bootstrapping to evaluate the statistical significance of the branching nodes/clustering?
Figure 2 represents the cluster obtained in the two-step method by applying Principal Component Analysis (PCA). In Figure 2A ten species-specific peaks (biomarkers) have been included and 5 biomarkers in Figure 2B. Bootstrapping or k-fold validation has been applied to all algorithms used in this study. It consisted of iterating the algorithms 10 times. In the case of PCA analysis, the clustering shown in Figure 2 was obtained after 10 iterations and for SVM and PLS-DA their validation is shown in Table S4. In Page 6, lines 216-219 we state “The validation of the method yielded a k-fold (k=10) score of 96.92% for PLS-DA performed over the peak matrix with 10 biomarkers and 98.46% for the analysis with 5 biomarkers. However, SVM algorithm achieved 100% discrimination in both cases when PCA was applied (Table S4).”
4) In addition to “hybrid HGM 2159”, two other hybrid isolates, “HGM 1657” and “HGM 4468”, also clearly cluster together with the other C. neoformans isolates in Figure 2B. However, in Figure 3, there appears to be only 1 hybrid isolate in the C. neoformans group.
In the two-step model, one interspecies hybrid clusters with C. neoformans in the dendrogram, built using 5 biomarkers and PCA. The model with 10 biomarkers clustered two more hybrids in the wrong group, which means that the PCA model with 5 biomarkers provided a better classification of the three Cryptococcus species. On the other hand, the application of SVM to the one-step model (Figure 3) allowed a similar classification to PCA in the two-step model (Figure 2B). Our point here is: SVM provides a good classification (only one interspecies hybrid misclassified) by the implementation of a much simpler, one-step, methodology.
5) Related to comment 4) above, in Figure 3, what are the criteria to divide the plot into the three areas (blue, green, and red) that represent different lineages? It seems arbitrary, and that perhaps “HGM 1657” and “HGM 4468” have been mistakenly grouped into the green area representing hybrids (based on the clustering in Figure 2B). Additionally, based on Figure 2B, the C. neoformans cluster that includes “hybrid HGM 2159” should represent a group that is more divergent from both the other hybrids as well as the rest of the C. neoformans. However, it appears that all of the C. neoformans isolates have been included in one group in Figure 3, which contradicts to the data in Figure 2B.
The lines dividing the three areas where most the isolates are correctly classified (except for one interspecies hybrid) are established by the SVM algorithm. They represent the “cut-off” values from the Principal Component 0 –PC0- (in the X-axis) and PC1 in the Y-axis that allow differentiation of the three Cryptococcus species. It is not arbitrary at all. On the contrary: we have performed different models that increasingly account for more Cryptococcus isolates correctly classified: the two-step model with 10 biomarkers (Figure 2A) misclassified 3 interspecies hybrid isolates, the same model misclassifies just one isolate (Figure 2B) and in a simplified, one-step method, all but one isolates –again- are correctly classified (Figure 3) and the validation of this model provides 100% reliability (Table S4). The robustness of our method is shown also in the fact that the 3 main C. neoformans clusters shown in the blue area (Figure 3) correlate with the 3 clusters in Figure 2B. Moreover, interspecies hybrid 2159 clusters with one of these C. neoformans clusters in exactly the same way as it does in Figure 2B. The closeness between some of the interspecies hybrids –in green- in Figure 3 and some C. neoformans isolates from the lower-left part of the blue area correlates with the closeness of these same clusters in Figure 2B. The same is true for C. deneoformans isolates, which are markedly distanced from the other two species, exactly the same way as in Figure 2B. In our opinion, there are no arbitrarities or contradictions between both figures. Figure 3 just represents a simpler, easier method to discriminate the three species in only one step instead of two.
Minor comments
- Maybe the “complex species” in the title should be “species complex”?
Yes. The title has been changed.
- Line 55. Remove the extra “VGII” right before “for gattii”.
Some characters had been missing in that line. It has been review and now it reads: “AFLP10/VGIV-VGIIIc for C. gattii complex”. Thank you for noticing this typo.
- Lines 83 and 84. The name of the same company is spelled differently.
The right spelling has been written (bioMérieux Marcy L'étoile, France)
- Lines 111 – 113. What is the classification for scores between 1.6 and 1.7, as in Table 1 and in line 182?
Score values within this category were usually considered as “low-confidence” identifications followed by the score in order to differentiate this category from the score values between 1.7 and 1.99. For the sake of clarity, in the manuscript this category is considered as “very low confidence identifications” but reliable since there is consistency along the top four identifications provided by MALDI-TOF. This has been detailed in Page 3, Lines 118-120: “Score values below 1.6 were only considered as “very low-confidence” identifications –but reliable- when consistent along the four top identifications…”
- Line 173. Should the “remaining neoformans isolates” be “remaining C. deneoformans isolates?”
No, that is correct. In the previous line we state: “correct identification of 18/22 C. neoformans isolates (81.8%)…” the remaining 4 isolates were unreliably identified with score values <1.6 (Table 1). That means that besides the low score, the top 4 identifications were not consistent. C. deneoformans isolates (n=3) were identified at the species level (score <1.6; n=1) and for the other two isolates MALDI-TOF only provided identification at the complex level (Table 1 legend).
Thank you for this thorough and constructive review.
- Reference No. 24. The “Cryptococcus neoformans” is missing in the title right before “species complex”.
That is right. “Cryptococcus neoformans” has been added again to the title.
Round 2
Reviewer 3 Report
In the revised manuscript, the authors addressed some of the comments raised by the reviewer. However, I think there are several major issues in the manuscript for which the authors have failed to deal with adequately.
- Related to the original major comment 3. The presentation of Table 1, as well as the related text describing the data within, are very confusing. For example, in the sentence in lines 184 – 185, at the beginning it says 2 isolates represent 8%, and then later on it says 13 isolates represent 29.5%. This is mathematically incorrect. Also, in the same paragraph, at the beginning it is talking about C. neoformans, then at the end, it says there are 8 isolates with scores below 1.6. However, if you look at Table 1, there are only 4 C. neoformans isolates with scores <1.6. Maybe the 8 isolates are referring to the “Total” number of isolates with <1.6 scores? I would strongly recommend the authors to go through the text carefully and make sure the information presented are accurate.
- Related to the original major comment 4. As pointed out in the original comment, there are discrepancies between results from Figure 2B and Figure 3. I understand the authors argue in their response that there is no discrepancy. I respectively disagree. For example, in line 212, the authors say there is “only one spectrum from interspecies hybrid misallocated in the C. neoformans cluster (Figure 2B).” However, as pointed out in the original comment, if you look at Figure 2B, there are two additional hybrids, “HGM 1657” and “HGM 4468” (the bottom two hybrids in the array) that are actually also located within the C. neoformans cluster. Just because these two isolates happen to be located next to the rest of the hybrids on the far right doesn’t mean they cluster together with the other hybrids. The topology of the tree clearly indicates that these two hybrids, together with the C. neoformans isolate HGM9890, form a cluster that belongs to the C. neoformans group. Additionally, the topology also suggests the bottom cluster that includes the hybrid isolate HGM2159 is more divergent when compared to the other C. neoformans and hybrid isolates, which is also contradicting to the conclusion from Figure 3 that all of the C. neoformans isolates belong to one single group. The authors need to revise the manuscript to reflect these differences between the two analyses and discuss it properly in Discussion.
- One of the caveats of the current study is the underrepresentation of C. deneoformans isolates. While I appreciate that the authors added a related sentence in the Discussion, I feel some additional discussion is warranted. For example, how does this bias affect the interpretation of the results, especially those characteristic peaks identified in Table 3?
- Some minor comments: 1) Line 84. For the company name, in one case there is a period before “Marcy” and in the other there is not? Which one is correct? 2) Line 85. It says all of the isolates were stored at -80 degree in water. Won’t this kill the cells? Why not storing them in glycerol, as standard lab protocol? 3) Lines 116-120. As in previous comments, what about the score between 1.6 and 1.7? This has not been defined. 4) Line 228. Maybe change “cross validation analysis was above” to “cross validation analysis and it was above”.
Author Response
In the revised manuscript, the authors addressed some of the comments raised by the reviewer. However, I think there are several major issues in the manuscript for which the authors have failed to deal with adequately.
- Related to the original major comment 3. The presentation of Table 1, as well as the related text describing the data within, are very confusing. For example, in the sentence in lines 184 – 185, at the beginning it says 2 isolates represent 8%, and then later on it says 13 isolates represent 29.5%. This is mathematically incorrect. Also, in the same paragraph, at the beginning it is talking about C. neoformans, then at the end, it says there are 8 isolates with scores below 1.6. However, if you look at Table 1, there are only 4 C. neoformans isolates with scores <1.6. Maybe the 8 isolates are referring to the “Total” number of isolates with <1.6 scores? I would strongly recommend the authors to go through the text carefully and make sure the information presented are accurate.
We appreciate the reviewer's comment. As the reviewer indicated, the rate of correct C. neoformans identifications at the species level was incorrect and has been recalculated (4.5%) as indicated above.
In the first version of the manuscript we mentioned the total amount of isolates with score values >1.6 -8- but, as the reviewer points out, this seemed to be confusing. Since total amounts were discussed in the previous paragraph, in lines 186-190 we specifically showed results related to C. neoformans identification.
The paragraph describing the score values obtained for the Cryptococcus species analyzed with the commercial database (Lines 186-190) had been modified: “Only two C. neoformans isolates (4.5%) were correctly identified at the species level with high-confidence score values (≥2.0) whilst 29.5% of the samples -13- were correctly identified with low-confidence scores (≥1.7) -Table 1-. Another 3 isolates were reliably identified to the species level, although with scores values ranging between 1.7 and 1.6 –very-low confidence identifications- and, finally, 4 isolates obtained scores below 1.6. The latter can be considered as unreliable identifications.”
We really do hope these results are clear now.
- Related to the original major comment 4. As pointed out in the original comment, there are discrepancies between results from Figure 2B and Figure 3. I understand the authors argue in their response that there is no discrepancy. I respectively disagree. For example, in line 212, the authors say there is “only one spectrum from interspecies hybrid misallocated in the C. neoformans cluster (Figure 2B).” However, as pointed out in the original comment, if you look at Figure 2B, there are two additional hybrids, “HGM 1657” and “HGM 4468” (the bottom two hybrids in the array) that are actually also located within the C. neoformans cluster. Just because these two isolates happen to be located next to the rest of the hybrids on the far right doesn’t mean they cluster together with the other hybrids. The topology of the tree clearly indicates that these two hybrids, together with the C. neoformans isolate HGM9890, form a cluster that belongs to the C. neoformans group.
Different approaches and algorithms were applied in order to correctly classify the three Cryptococcus species. Although the model is not perfect, the improvements in this task can be seen when Figure 2A, 2B and 3 are closely analyzed. There are no discrepancies among them but an increasingly accurate way to discriminate the three species. Figure 2A represents the clustering of these species when PCA is applied to a matrix where 10 discriminating biomarkers have been included. Six isolates are misclassified with this model. Therefore, it was improved by including the 5 biomarkers that provided higher discrimination. Results can be seen in Figure 2B, with 3 –not one, as the reviewer remarked- hybrid isolates misclassified. In our effort to still improve the classification obtained by PCA with 5 biomarkers –Figure 2B- SVM was applied to these same biomarkers. As shown in Figure 3, this model correctly classified 98.46% of the isolates –Page 6, Lines 228-229-.
We agree with the reviewer that the interspecies hybrids HGM1657 and 4468 are misallocated in a cluster with C. neoformans. This has been explained in the manuscript in Pages 5-6, Lines 214-215: “…with only three spectra from interspecies hybrids misallocated in the C. neoformans cluster (Figure 2B).”
Additionally, the topology also suggests the bottom cluster that includes the hybrid isolate HGM2159 is more divergent when compared to the other C. neoformans and hybrid isolates, which is also contradicting to the conclusion from Figure 3 that all of the C. neoformans isolates belong to one single group. The authors need to revise the manuscript to reflect these differences between the two analyses and discuss it properly in Discussion.
The diversity of the C. neoformans isolates included in the study was expected due to the long period of time in which they were collected (1994-2007) and the variety of samples were they were found. This diversity is represented not only in Figure 2B, as the reviewer points out, but also in Figure 3, where at least 3 major clusters (represented in the upper, medium and lower part of the blue area) can be seen. But all of them –except for one, as shown in Figure 3- can be differentiated from the interspecies hybrids, the phylogenetically closest Cryptococcus species. Besides, the correct identification of these diverse isolates both by the in-house library and the developed models is one of the strengths of the study and shows the robustness of both approaches.
This fact has been revised and discuss in Page 6, Lines 216-218 “Besides, the variability of C. neoformans was shown by the three main clusters in which the protein spectra from this species grouped (Figure 2B)” and Page 10, Lines 350-352: “This algorithm allowed correct discrimination of all C. deneoformans isolates and of a high rate of C. neoformans and interspecies hybrids despite their genetic variability, demonstrating the robustness of this study.”
- One of the caveats of the current study is the underrepresentation of C. deneoformans isolates. While I appreciate that the authors added a related sentence in the Discussion, I feel some additional discussion is warranted. For example, how does this bias affect the interpretation of the results, especially those characteristic peaks identified in Table 3?
Both the identification of C. deneoformans using the expanded, in-house library and by peak analysis allowed the correct differentiation of this species which, as shown in Figure 2B and 3, is phylogenetically different from the other two Cryptococcus species. The two species that showed higher homology are C. neoformans and the interspecies hybrids, which are well represented in the study.
Although the number of C. deneoformans in the study is obviously limited, its identification with the in-house library was reliable. Although other species-specific peaks for C. deneoformans could be found in further studies, those found in this study (6576.08, 7103.01 and 7830.42 m/z) are not present the other two Cryptococcus species and allow correct discrimination of C. deneoformans so far.
The following text was included in the manuscript in order to clarify this part (Page 10, Lines 323-326): “Although the limited number of C. deneoformans included in this study is clearly one of its limitations, the fact that all isolates from this species were reliably identified at the species level with the expanded library shows that even small improvements in the reference library were useful to achieve correct identifications of this species.” and (Page 10, Lines 339-341): “The implementation of PLS-DA analysis in a two-step approach allowed the discrimination of C. deneoformans isolates in the first place due to the phylogenetic differences between this species and the other two Cryptococcus species.”
- Some minor comments:
1) Line 84. For the company name, in one case there is a period before “Marcy” and in the other there is not? Which one is correct?
The correct name of the company and the location of its headquarters (bioMérieux, Marcy-l'Étoile, France) has been reviewed and correctly written
2) Line 85. It says all of the isolates were stored at -80 degree in water. Won’t this kill the cells? Why not storing them in glycerol, as standard lab protocol?
Storing the isolates in water at -80ºC has been a usual practice in our hospital over the last 25 years and we do not usually have problems with the viability of the isolates. To avoid the loss of isolates we have two copies of each isolate. However, the Cryptococcus isolates included in this study had been kept for over 20 years using this method and only 5 isolates –frozen and thawed over time- could not be recovered from the frozen back up, probably because they were very old and this species is less viable than other yeast species.
3) Lines 116-120. As in previous comments, what about the score between 1.6 and 1.7? This has not been defined.
The definition of the identifications with score values between 1.6 and 1.7 is stated in lines 120-121: “Score values below 1.6 were only considered as “very low-confidence” identifications –but reliable- when consistent along the four top identifications, otherwise they were considered as “not reliable”.”
4) Line 228. Maybe change “cross validation analysis was above” to “cross validation analysis and it was above”.
Agree.
Round 3
Reviewer 3 Report
Line 20. “Time-of” should be “- Time of”. Line 23. “close-related” should be “closely-related”. Line 29. There is an extra “isolates” after “Cryptococcus”. Line 119-120. Maybe change “below 1.6” to “below 1.7” to keep it consistent with descriptions in line 189-190.
This manuscript is a resubmission of an earlier submission. The following is a list of the peer review reports and author responses from that submission.
Round 1
Reviewer 1 Report
The authors present a study in which they used an already published method to identify Cryptococcus spp. that have already been identified previously. The only result that maybe is kind of new to the readers is the identification of the Cryptococcus neoformans complex. But even this has been shown to be possible before. Currently, the reviewer does not see the novelty of the study. Furthermore, the manuscript is not well structured, e.g. "2.2 Molecular identification" includes the part about MSP creation for MALDI-TOF MS. References are missing for the extraction methods and so on.
The abstract is confusing and I am not sure if the picture of the MALDI-TOF MS in Fig. 1 is owned by the authors since the reviewer thinks to have seen this picture already on a Bruker advertisement. But maybe he is wrong.
Finally, all genus and species names are not written in italic. Figure 2 is not readable. Figure 3 is blurred. The conclusion is not supported by the study. The authors only showed the usefulness of the method for Cryptococcus species but state to have shown the usefulness for all non-Candida yeast. The authors did not test other non-Candida yeast.
Overall, the manuscript needs to be rewritten and the aim of the study must be clear. The title is misleading since the new thing about this study is not identifying Cryptococcus spp. (which has been shown many times before) but showing the usefulness of MALDI-TOF MS by identifying C. neoformans complex. Even this seems to be done in some way already before with MALDI-TOF MS.
The reviewer thinks the authors should thoughtfully reorganize the manuscript.
Make sure that it becomes clear what is really new about their study and why this is of major importance to the research world. Formal things, such as Italic and proper Figures should be basic things that are not worth discussing. In the current state, the reviewer does not think it useful if further comments on the manuscript are added until these basic things and the overall manuscript is improved.
Reviewer 2 Report
The paper presented by Zvezdanova and co-workers presents the application of Matrix-Assisted Laser Desorption/Ionization Time-of Flight Mass Spectrometry and peak analysis in the microbiology analysis of Cryptococcus species identification. The presented study may be useful for such kind of study however, the manuscript, in my opinion, requires further improvements.
There is a lot of questions and comments which should be explained by the authors.
My comments are presented below.
Major concerns:
- Abstract – clearly demonstrate the importance of the presented topic,
- Abstract – the names of species should be presented in Italics,
- Abstract – All abbreviations should be explained,
- Abstract – why do the authors think that the presented method is cost-efficient? Please explain.
- All manuscript – clearly present the units, especially page 3, lines 107-109,
- What criteria do you used for in-house database formation? Please explain.
- Introduction – explain why this topic is important, why MALDI may be used for such investigation,
- Materials and Methods – Isolates – how the isolates were stored between the presented years?
- Materials and Methods – MALDI-TOF – page 3, line 101 – define city and country for Bruker,
- Results - page 4, line 168 – neoformans isolates –n=4- - it should be – n=4 -?
- Results, Table 1 – the names of species are presented in Italics, the same should be in all of the text,
- Figure 2 – the quality of the presented figure is low and should be improved,
- References – check the reference format according to the instructions for authors;
- Explain what kind of proteins did you identify, show MALDI-TOF spectra,
- The manuscript is available in Preprints page and researchgate – explain why?
The presented manuscript should better present the importance of the presented study and the applicability of the proposed method. The results should be better presented. In this form it cannot be accepted for publication.